# Attacks Meet Interpretability:
# Attribute-steered Detection of Adversarial Samples

**Guanhong Tao,*  Shiqing Ma,*  Yingqi Liu, Xiangyu Zhang**
Department of Computer Science, Purdue University
{taog, ma229, liu1751, xyzhang}@cs.purdue.edu

## Abstract

Adversarial sample attacks perturb benign inputs to induce DNN misbehaviors. Recent research has demonstrated the widespread presence and the devastating consequences of such attacks. Existing defense techniques either assume prior knowledge of specific attacks or may not work well on complex models due to their underlying assumptions. We argue that adversarial sample attacks are deeply entangled with interpretability of DNN models: while classification results on benign inputs can be reasoned based on the human perceptible features/attributes, results on adversarial samples can hardly be explained. Therefore, we propose a novel adversarial sample detection technique for face recognition models, based on interpretability. It features a novel bi-directional correspondence inference between attributes and internal neurons to identify neurons critical for individual attributes. The activation values of critical neurons are enhanced to amplify the reasoning part of the computation and the values of other neurons are weakened to suppress the uninterpretable part. The classification results after such transformation are compared with those of the original model to detect adversaries. Results show that our technique can achieve 94% detection accuracy for 7 different kinds of attacks with 9.91% false positives on benign inputs. In contrast, a state-of-the-art feature squeezing technique can only achieve 55% accuracy with 23.3% false positives.

## 1  Introduction

Deep neural networks (DNNs) have achieved great success in a wide range of applications such as natural language processing [1], scene recognition [2], objection detection [3], anomaly detection [4], etc. Despite their success, the wide adoption of DNNs in real world missions is hindered by the security concerns of DNNs. It was shown in [5] that slight imperceptible perturbations are capable of causing incorrect behaviors of DNNs. Since then, various attacks on DNNs have been demonstrated [6–9]. Detailed discussion of these attacks can be found in §2. The consequences could be devastating, e.g., an adversary could use a crafted road sign to divert an auto-driving car to go offtrack [10, 11]; a human uninterpretable picture may fool a face ID system to unlock a smart device.

Existing defense techniques either harden a DNN model so that it becomes less vulnerable to adversarial samples [5, 6, 12–14] or detect such samples during operation [15–17]. However, many these techniques work by additionally training on adversarial samples [5, 6, 12], and hence require prior knowledge of possible attacks. A state-of-the-art detection technique, *feature squeezing* [18], reduces the feature space (e.g., by reducing color depth and smoothing images) so that the features that adversaries rely on are corrupted. However, according to our experiment in §4.2, since the technique may squeeze features that the normal functionalities of a DNN depend on, it leads to degradation of accuracy on both detecting adversarial samples and classifying benign inputs.

Our hypothesis is that adversarial sample attacks are deeply entangled with interpretability of DNN classification outputs. Specifically, in a fully connected network, all neurons are directly/transitively influencing the final output. Many neurons denote abstract features/attributes that are difficult for humans to perceive. An adversarial sample hence achieves its goal by leveraging these neurons. As

---

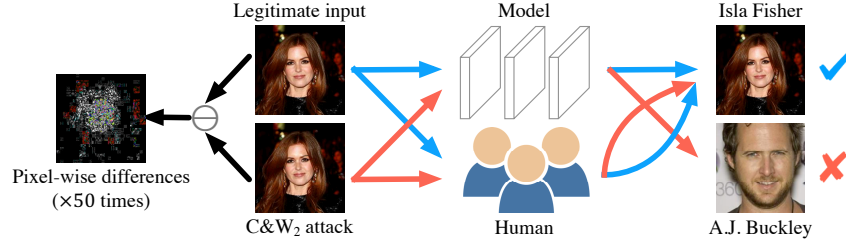

Figure 1: Comparison of DNNs and humans in face recognition. Pixel-wise differences have been magnified 50 times for better visual display. The left two images of actress Isla Fisher are the inputs for a DNN model and human to recognize. The rightmost two images are the representative images of the corresponding identities. Blue arrows illustrate the paths of the benign input. Red arrows denote the paths of the adversarial sample generated using the C&W $L_2$ attack [8].

such, while the classification result of a benign input is likely based on a set of human perceptible attributes (e.g., a given face picture is recognized as a person $A$ because of the color of eyes and the shape of nose), the result of an adversarial input can hardly be attributed to human perceptible features. That is, the DNN is not utilizing interpretable features, or *guessing*, for adversarial samples. Figure 1 illustrates a sample attack. The adversarial sample is generated using the method proposed by Carlini and Wagner [8] (C&W). Observe that it is almost identical to the original benign input. The DNN model demonstrated in Figure 1 is VGG-Face [19], one of the most well-known and highly accurate face recognition systems. In Figure 1, a picture of actress Isla Fisher with adversarial perturbations is incorrectly recognized as actor A.J. Buckley. Apparently, the classification of the adversarial sample must not depend on the human perceptible attributes such as eyes, nose, and mouth as these attributes are (almost) identical to those in the original image. In fact, humans would hardly misclassify the perturbed image. The overarching idea of our technique is hence to *inspect if the DNN produces its classification result mainly based on human perceptible attributes. If not, the result cannot be trusted and the input is considered adversarial.*

We propose an adversarial sample detection technique called AmI (Attacks meet Interpretability) for face recognition systems (FRSes) based on interpretability. FRSes are a representative kind of DNNs that are widely used. Specifically, AmI first extracts a set of neurons (called *attribute witnesses*) that are critical to individual human face attributes (e.g., eyes and nose) leveraging a small set of training images. While existing DNN model interpretation techniques can generate input regions that maximize the activation values of given neurons [20–24] or identify neurons that are influenced by objects (in an image), these techniques are insufficient for our purpose as they identify weak correlations. AmI features a novel reasoning technique that discloses strong correlations (similar to one-to-one correspondence) between a human face attribute and a set of internal neurons. It requires bi-directional relations between the two: *attribute changes lead to neuron changes* and *neuron changes imply attribute changes*, whereas most existing techniques focus on one direction relations. After identifying attribute witnesses, a new model is constructed by transforming the original model: strengthening the values of witness neurons and weakening the values of non-witness neurons. Intuitively, the former is to enhance the reasoning aspect of the FRS decision making procedure and the latter is to suppress the unexplainable aspect. Given a test input, inconsistent prediction results by the two models indicate the input is adversarial. In summary, we make the following contributions:

- We propose to detect adversarial samples by leveraging that the results on these samples can hardly be explained using human perceptible attributes.
- We propose a novel technique AmI that utilizes interpretability of FRSes to detect adversarial samples. It features: (1) bi-directional reasoning of correspondence between face attributes and neurons; (2) attribute level mutation instead of pixel level mutation during correspondence inference (e.g., replacing the entire nose at a time); and (3) neuron strengthening and weakening.
- We apply AmI to VGG-Face [19], a widely used FRS, with 7 different types of attacks. AmI can successfully detect adversarial samples with true positive rate of 94% on average, whereas a state-of-the-art technique called feature squeezing [18] can only achieve accuracy of 55%. AmI has 9.91% false positives (i.e., misclassifies a benign input as malicious) whereas feature squeezing has 23.32%.
- AmI is available at GitHub [25].

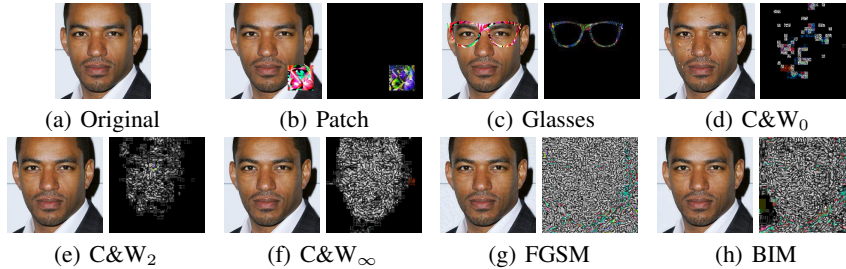

(a) Original       (b) Patch       (c) Glasses       (d) C&W$_0$

(e) C&W$_2$       (f) C&W$_\infty$       (g) FGSM       (h) BIM

Figure 2: Adversarial samples of different attacks on FRSes. Figure 2(a) shows the original image of actor Laz Alonso. Figure 2(b)-(h) are 7 adversarial samples generated using different attacks (denoted in the caption). The left images are the generated adversarial samples and the right images are their pixel-wise differences compared to the original image. The differences have been magnified 50 times for better visual display.

## 2   Background and Related Work

• **Model Interpretation:** Interpreting DNNs is a long-standing challenge. Existing work investigates interpretability from multiple perspectives and feature visualization is one of them.While explaining DNNs largely lies in understanding the role of individual neurons and layers, feature visualization tackles the problem by generating neurons' *receptive fields*, which are input regions of interest maximizing the activation value of a neuron [20–24]. In [26], the authors proposed a technique to associate neurons with objects in an input image. Particularly, if the receptive field of a set of neurons overlaps with some human-interpretable object (e.g., dog and house), the set of neurons is labeled with the corresponding object and regarded as the *detector of the object*. This approach expands the focus from one neuron (in feature visualization) to a set of neurons. However, while these existing techniques are related to attribute witness extraction in our technique, they are not sufficient for our purpose. The reasoning in feature visualization and in [26] is one-way, that is, pixel perturbations lead to neuron value changes. In §3.1 and §4.2, we discuss/demonstrate that one-way reasoning often leads to over-approximation (i.e., the neurons are related but not essential to some attribute) and hence sub-optimal detection results. A stronger notion of correlation is needed in our context. Furthermore, we leverage attribute substitutions in the training images for mutation (e.g., replacing a nose with other different noses), instead of using random pixel perturbations as in existing work.

• **Adversarial Samples:** Adversarial samples are model inputs generated by adversaries to fool DNNs. There exist two types of adversarial sample attacks: *targeted attacks* and *untargeted attacks*. Targeted attacks manipulate DNNs to output a specific classification result and hence are more catastrophic. Untargeted attacks, on the other hand, only lead to misclassification without a target.

*Targeted attacks* further fall into two categories: *patching* and *pervasive perturbations*. The former generates patches that can be applied to an image to fool a DNN [10, 27]. The left image of Figure 2(b) shows an example of patching attack. Figure 2(c) demonstrates a patch in the shape of glasses [28]. In contrast, pervasive perturbation mutates a benign input at arbitrary places to construct adversarial samples. Carlini and Wagner [8] proposed three such attacks. Figure 2(d) shows the $L_0$ attack that limits the number of pixels that can be altered without the restriction on their magnitude; Figure 2(e) shows the $L_2$ attack that minimizes the Euclidean distance between adversarial samples and the original images. Figure 2(f) demonstrates the $L_\infty$ attack, which bounds the magnitude of changes of individual pixels but does not limit the number of changed pixels.

*Untargeted attacks* was first proposed by Szegedy et al. [5]. Fast Gradient Sign Method (FGSM) generates adversarial samples by mutating the whole image in one step along the gradient direction [6] (Figure 2(g)). Kurakin et al. [7] subsequently extended FGSM by taking multiple small steps. The attack is called Basic Iterative Method (BIM), which is shown in Figure 2(h).

• **Defense Techniques:** Researchers have made a plethora of efforts to defend DNNs against adversarial attacks. Existing defense techniques can be broadly categorized into two kinds: *model hardening* and *adversary detection*. Our proposed AmI system lies in the second genre.

*Model hardening* is to improve the robustness of DNNs, which is to prevent adversarial samples from causing DNN misbehaviors. Previous work regarded adversarial samples as an additional output class and trained DNNs with the new class [5, 6, 12]. These approaches, however, may not be practical due to the requirement of prior knowledge of adversarial samples. Defensive distillation [14] aimed

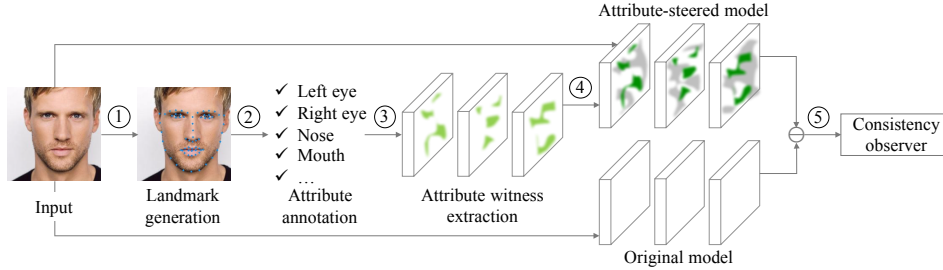

Figure 3: Architecture of AmI. It first identifies shapes of human face attributes (step ①) and then requests humans to annotate the identified shapes (step ②). For each attribute, AmI automatically extracts *attribute witnesses* in the original model (step ③). Extracted *attribute witnesses* are subsequently leveraged to construct an attribute-steered model (step ④). For a test input, the new model along with the original model are executed side-by-side to detect output inconsistencies, which indicate adversarial samples (step ⑤).

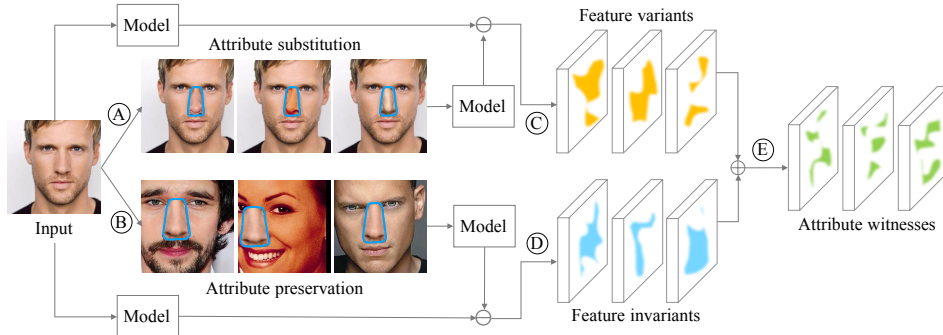

Figure 4: Attribute witness extraction. *Attribute substitution* (step Ⓐ) is applied on a base image by substituting an attribute with the counterpart in other images and then observing the neurons that have different values (step Ⓒ). Observe that the images in the top row have different noses but the same face. *Attribute preservation* (step Ⓑ) preserves an attribute by replacing the attribute in other images with that of the base and then observing the neurons that do not change (step Ⓓ). Observe that the different faces in the bottom row have the same nose (copied from the base). The two sets are intersected to yield *attribute witnesses* (step Ⓔ).

to hide gradient information from adversaries. This strategy was implemented through replacing the last layer with a modified softmax function.

*Adversary detection* identifies adversarial samples during execution [15–17]. Grosse et al. [16] used a statistical test to measure distance between adversarial samples and benign inputs. This technique requires adversarial samples in advance, which may not be satisfied in practice. Xu et al. [18], on the other hand, proposed a feature squeezing strategy that does not require adversarial samples. It shrinks the feature space by reducing color depth from the 8-bit scale to smaller ones and by image smoothing. After feature squeezing, adversarial samples are likely to induce different classification results (compared to without-squeezing), whereas benign inputs are not. It is considered the state-of-the-art approach to detecting adversarial samples. Our AmI system also relies on classification inconsistencies, although AmI leverages interpretability of FRSes and has significantly better performance than feature squeezing in the context of face recognition, as shown in §4.2.

## 3  Approach

We aim to explore using interpretability of DNNs to detect adversarial samples. We focus on FRSes, whose goal is to recognize identities of given human face images. In this specific context, the overarching idea of our technique is to determine if the prediction results of a FRS DNN are based on human recognizable features/attributes (e.g., eyes, nose, and mouth). Otherwise, we consider the input image adversarial. Our technique consists of two key steps: (1) identifying neurons that correspond to human perceptible attributes, called *attribute witnesses*, by analyzing the FRS's behavior on a small set of training inputs; (2) given a test image, observe the activation values of attribute witnesses and their comparison with the activation values of other neurons, to predict if the image is adversarial. In (2), transformation of activation values are also applied to enhance the contrast.

The workflow of AmI is shown in Figure 3. Firstly, we analyze how a FRS recognizes an input image using human perceptible attributes to extract attribute witnesses. Steps ①-③ in Figure 3 illustrate the extraction process. For a training input, we detect shapes of face attributes leveraging an automatic technique dlib [29] in step ①. Observe that after this step, the important shapes are identified by dotted lines. In step ②, human efforts are leveraged to annotate shapes with attributes. While automation could be achieved using human face attribute recognition DNN models, the inherent errors in those models may hinder proving the concept of our technique and hence we decided to use human efforts for this step. Note that these are one-time efforts and the annotated images can be reused for multiple models. For each human identified attribute, we extract the corresponding neurons in the FRS (step ③), which are the witness of the attribute. This step involves a novel bi-directional reasoning of the correlations between attribute variations and neuron variations. Details are elaborated in §3.1. For each attribute, we apply these steps to a set of 10 training inputs (empirically sufficient) to extract the statistically significant witnesses. Secondly, we leverage the extracted witnesses to construct a new model for adversarial sample detection. Steps ④-⑤ in Figure 3 show the procedure. In step ④, we acquire a new model by strengthening the witness neurons and weakening the other neurons. We call the resulted model an *attribute-steered model*. Intuitively, the new model aims to suppress the uninterpretable parts of the model execution and encourage the interpretable parts (i.e., the *reasoning parts*). This is analogous to the retrospecting and then reinforcing process in human decision making. Details are discussed in §3.2. To detect adversarial samples, for each test input, we use both the original model and the attribute-steered model to predict its identity. If inconsistent predictions are observed (step ⑤), we mark it adversarial.

## 3.1 Attribute Witness Extraction

The first step of AmI is to determine the *attribute witnesses*, i.e., the neurons corresponding to individual attributes. The extraction process is illustrated in Figure 4, which consists of reasonings in two opposite directions: *attribute substitution* (step Ⓐ) and *attribute preservation* (step Ⓑ). More specifically, in step Ⓐ, given an input image called *the base image* and an attribute of interest, our technique automatically substitutes the attribute in the image with the corresponding attribute in other images. For instance in the first row of images in Figure 4, the nose in the original image is replaced with different other noses. Subsequently, the base image and each substituted image are executed by the FRS and the internal neurons that have non-trivial activation value differences are extracted as a superset of the witness of the replaced attribute (step Ⓒ). However, not all of these neurons are essential to the attribute. Hence, we further refine the neuron set by *attribute preservation*. Particularly in step Ⓑ, we use the attribute from a base image to replace that in other images. We call the resulted images the attribute-preserved images, which are then used to extract the neurons whose activation values have no or small differences compared to those of the base image in step Ⓓ. Intuitively, since the same attribute is retained in all images, the witness neurons shall not have activation value changes. However, the neurons without changes are not necessarily essential to the attribute as some abstract features may not change either. Therefore, we take intersection of the neurons extracted in *attribute substitution* and *attribute perservation* as the witnesses (step Ⓔ).

The design choice of bi-directional reasoning is critical. Ideally, we would like to extract neurons that have strong correlations to individual attributes. In other words, given an attribute, its witness would include the neurons that have correspondence with the attribute (or are kind of "*equivalent to*" the attribute). Attribute substitution denotes the following *forward reasoning*.

$$\textit{Attribute changes} \longrightarrow \textit{Neuron activation changes} \quad\quad (1)$$

However, this one-way relation is weaker than equivalence. Ideally, we would like the witness neurons to satisfy the following *backward reasoning* as well.

$$\textit{Neuron activation changes} \longrightarrow \textit{Attribute changes} \quad\quad (2)$$

However such backward reasoning is difficult to achieve in practice. We hence resort to reasoning the following property that is **logically** equivalent to Equation 2 but forward computable.

$$\textit{No attribute changes} \longrightarrow \textit{No neuron activation changes} \quad\quad (3)$$

The reasoning denoted by Equation 3 essentially corresponds to attribute preservation. In §4.2, we show that one-way reasoning produces inferior detection results than bi-directional reasoning.

**Detailed Design.** Suppose $F : X \rightarrow Y$ denotes a FRS, where $X$ is the set of input images and $Y$ is the set of output identities. Let $f^l$ represent the $l$-th layer and $x \in X$ denote some input. A FRS with $n + 1$ layers can be defined as follows.

$$F(x) = \text{softmax} \circ f^n \circ \cdots \circ f^1(x). \tag{4}$$

For each layer $f^l$, it takes a vector of activations from the previous layer and yields a vector of new activations. Each new activation value (denoting a neuron in the layer) is computed by a neuron function that takes the activation vector of the previous layer and produces the activation value of the neuron. In other words,

$$f^l(p) = \langle f_1^l(p), \cdots, f_m^l(p) \rangle, \tag{5}$$

where $m$ the number of neurons at layer $l$, $p$ the activation vector of layer $l - 1$, and $f_j^l(p)$ the function of neuron $j$.

*Attribute substitution* reasons that attribute changes lead to neuron value changes. Specifically, it replaces an attribute (e.g., nose) in a base image with its counterpart in other images, and then observes neuron variations. The activation difference of neuron $j$ at layer $l$ is defined as follows.

$$\Delta f_{j,as}^l = |f_j^l(p) - f_j^l(p_{as})|, \tag{6}$$

where the subscript $as$ denotes the substituted image. The magnitude of variations indicates the strength of *forward relations* between the attribute and the corresponding neurons. Larger variation indicates stronger relations. Specifically, we consider a neuron $f_j^l(p)$ a candidate (of witness) if its variation is larger than the median of all the observed variations in the layer. Let $g_{as}^{i,l}(p)$ be the set of witness candidates for attribute $i$ at layer $l$. It can be computed as follows.

$$g_{as}^{i,l}(p) = \{f_j^l(p) \mid \Delta f_{j,as}^l > median_{j \in [1,m]}(\Delta f_{j,as}^l)\} \tag{7}$$

We compute $g_{as}^{i,l}(p)$ for 10 training images and take a majority vote. Note that witnesses denote properties of a (trained) FRS which can be extracted by profiling a small number of training runs.

It is worth pointing out that a key design choice is to replace an attribute with its counterparts (from other persons). A simple image transformation that eliminates an attribute (e.g., replacing nose with a black/white rectangle) does not work well. Intuitively, we are reasoning about how the different instantiations of an attribute (e.g., noses with different shapes and different skin colors) affect inner neurons. However, eliminating the attribute instead reasons about how the presence (or absence) of the attribute affects neurons, which is not supposed to be a basis for identity recognition.

*Attribute preservation* reasons that no changes of an attribute lead to no changes of its witness neurons. In other words, it retains the attribute of the base image by copying it to other images. Let $g_{ap}^{i,l}(p)$ represent the set of neuron functions that do not vary much under the transformation for attribute $i$ at layer $l$, with the subscript $ap$ representing attribute preservation. It can be computed as follows.

$$g_{ap}^{i,l}(p) = \{f_j^l(p) \mid \Delta f_{j,ap}^l \leq median_{j \in [1,m]}(\Delta f_{j,ap}^l)\}, \tag{8}$$

It essentially captures the neurons that have strong *backward relations* (Equation 2 and Equation 3).

The witness $g^i$ for attribute $i$ is then computed by intersection as follows.

$$g^i = \bigcup_{l=1}^{n} g^{i,l}(p) = \bigcup_{l=1}^{n} g_{as}^{i,l}(p) \cap g_{ap}^{i,l}(p) \tag{9}$$

At the end, we want to point out that the witness sets of different attributes may overlap, especially for neurons that model correlations across attributes (see Table 1).

## 3.2 Attribute-steered Model

After witnesses are extracted, a new model is constructed by transforming the original model (without additional training). We call it the *attribute-steered model*. Particularly, we strengthen the witness neurons and weaken the others. Intuitively, we are emphasizing reasoning and suppressing gut-feelings. For a benign input that the FRS can reason about based on attributes, the new model would be able to produce consistent prediction result as the original model. In contrast for an adversarial input, the new model would produce inconsistent result since such an input can hardly be reasoned.

*Neuron weakening* reduces the values of non-witness neurons in each layer that have activation values larger than the mean of the values of witness neurons in the layer. The following reduction is

performed on all the satisfying (non-witness) neurons.

$$v' = e^{-\frac{v-\mu}{\alpha \cdot \sigma}} \cdot v, \tag{10}$$

where $v$ denotes the value of a non-witness neuron; $\mu$ and $\sigma$ are the mean and standard deviation of values of witness neurons in the same layer, respectively; $\alpha$ defines the magnitude of weakening, which is set to 100 in this paper. Note that the larger values the non-witness neurons have, the more substantial reduction they undertake. The function $e^{-x}$ employed here is inspired by the Gaussian function $e^{-x^2}$ widely used as a noise-removing kernel in image processing [30–32].

*Neuron strengthening* enlarges the values of all witness neurons as follows.

$$v' = \epsilon \cdot v + \left(1 - e^{-\frac{v-\min}{\beta \cdot \sigma}}\right) \cdot v, \tag{11}$$

where $\epsilon$ is the strengthening factor, and the second term ranges in $[0, v)$ (since $v - \min$ must be great than 0). The presence of the second term allows the larger activation values to have more substantial scale-up (more than just $\epsilon$). $\epsilon$ and $\beta$ are set to $1.15$ and $60$, respectively in this paper. They are chosen through a tuning set of 100 benign images, which has no overlap with the test set. Weakening and strengthening are applied on each test input during production run layer by layer.

In addition to weakening and strengthening, we also apply attribute conserving transformation to further corrupt the uninterpretable features leveraged by adversarial samples. The activations of a convolution/pooling layer are in the form of matrix. In this transformation, we remove the margin of matrix and then reshape it back to the original size through interpolation. Note that the attribute-steered model is resilient to attribute conserving transformations for benign inputs whose prediction results are based on attributes. In particular, for non-witness neurons in pooling layers, we resize them by first removing the margins and then enlarging the remaining to fit the layers. For instance, the shape of activations in pool3 layer of VGG-Face [19] is $28 \times 28$. We first remove the margin with size of 2 and get a new $24 \times 24$ shape of activations. And then we use a bicubic interpolation to resize the activations back to $28 \times 28$. As the transformation is only applied to non-witness neurons, it is regarded as part of neuron weakening, whose results are presented in the WKN row of Table 3.

## 4   Experiments

We use one of the most widely used FRSes, VGG-Face [19] to demonstrate effectiveness of AmI. Three datasets, VGG Face dataset (VF) [18], Labeled Faces in the Wild (LFW) [33] and CelebFaces Attributes dataset (CelebA) [34] are employed. We use a small subset of the VF dataset (10 images) to extract attribute witnesses, which has no intersection with other datasets used in testing for evaluating both the quality of extracted attribute witnesses and adversarial sample detection. Since witnesses denote properties of a trained FRS, their extraction only requires profiling on a small set of benign inputs. The experimental results are compared to a state-of-the-art detection technique, feature squeezing [18] on seven kinds of adversarial attacks. AmI is publicly available at GitHub [25].

### 4.1   Evaluation of Extracted Attribute Witnesses

The witnesses extracted can be found in Table 1, which shows the layers (1st row), the number of neurons in each layer (2nd row) and the number of witnesses (remaining rows). Note that although there are 64-4096 neurons in each layer, the number of witnesses extracted is smaller than 20. Some layers do not have extracted witnesses due to their focus on fusing features instead of abstracting features. To evaluate the quality of the extracted witnesses, that is, how well they describe the corresponding attribute, we design the following experiment similar to that in [35]. For each attribute, we create a set of images that may or may not contain the attribute and then we train a two-class model to predict the presence of the attribute based on the activation values of the witnesses. The prediction accuracy indicates the quality of the witnesses. The model has two layers. The first layer takes the activation values of the witness neurons of the attribute (across all layers). The second layer is the output layer with two output classes. We use 2000 training images from the VF set (1000 with the attribute and 1000 without the attribute) to train the model. Then we test on a disjoint set of 400 test images from both VF and LFW (200 with the attribute and 200 without the attribute). For comparison, we also use the activation values of the *face descriptor* layer of VGG-Face (i.e., fc7 layer) to predict presence of attributes. This layer was considered by other researchers as representing abstract human face features in [19]. The results are shown in Table 2. For the VF dataset, witnesses

Table 1: Number of extracted attribute witnesses in VGG-Face. The first row lists the layers used in VGG-Face. The second row shows the number of neurons at each layer. The following four rows present the number of extracted witnesses for individual attributes. The following six rows denote the pairwise overlap of witnesses between different attributes. The bottom row shows the number of neurons shared by the witness sets of different attributes.

| Layer Name | conv1_1 | conv1_2 | pool1 | conv2_1 | conv2_2 | pool2 | conv3_1 | conv3_2 | conv3_3 | pool3 |
|---|---|---|---|---|---|---|---|---|---|---|
| #Neuron | 64 | 64 | 64 | 128 | 128 | 128 | 256 | 256 | 256 | 256 |
| #Left Eye | 1 | - | - | - | 2 | 3 | 4 | 2 | 3 | 2 |
| #Right Eye | 1 | - | - | - | 3 | 3 | 4 | 3 | 2 | 3 |
| #Nose | 1 | - | - | - | 1 | 3 | 2 | - | 1 | 3 |
| #Mouth | 1 | - | - | - | 3 | 2 | 4 | 3 | 15 | 7 |
| #Left Eye   & Right Eye | 1 | - | - | - | 2 | 3 | 3 | 1 | - | - |
| #Left Eye   & Nose | 1 | - | - | - | 1 | 3 | 2 | - | - | - |
| #Left Eye   & Mouth | 1 | - | - | - | 2 | 1 | 2 | 1 | 1 | - |
| #Right Eye & Nose | 1 | - | - | - | 1 | 3 | 1 | - | - | - |
| #Right Eye & Mouth | 1 | - | - | - | 3 | 1 | 2 | 2 | 1 | 1 |
| #Nose        & Mouth | 1 | - | - | - | 1 | 1 | 1 | - | - | - |
| #Shared | 1 | - | - | - | 1 | 1 | 1 | - | - | - |

| Layer Name | conv4_1 | conv4_2 | conv4_3 | pool4 | conv5_1 | conv5_2 | conv5_3 | pool5 | fc6 | fc7 |
|---|---|---|---|---|---|---|---|---|---|---|
| #Neuron | 512 | 512 | 512 | 512 | 512 | 512 | 512 | 512 | 4096 | 4096 |
| #Left Eye | 9 | 5 | 15 | 7 | 12 | 4 | 1 | 1 | - | 1 |
| #Right Eye | 7 | 3 | 10 | 9 | 9 | 1 | - | - | - | - |
| #Nose | 10 | 8 | 17 | 13 | 7 | 2 | 2 | 1 | - | 1 |
| #Mouth | 19 | 12 | 12 | 11 | 8 | 2 | 1 | 2 | 1 | 1 |
| #Left Eye   & Right Eye | 5 | 1 | 3 | 4 | 2 | - | - | - | - | - |
| #Left Eye   & Nose | 3 | - | 4 | - | 1 | - | - | - | - | - |
| #Left Eye   & Mouth | 1 | 1 | - | - | - | - | - | - | - | - |
| #Right Eye & Nose | 3 | - | 1 | 1 | 1 | - | - | - | - | - |
| #Right Eye & Mouth | 2 | - | 2 | - | - | - | - | - | - | - |
| #Nose        & Mouth | 5 | 1 | 2 | 2 | - | - | - | - | - | - |
| #Shared | 1 | - | - | - | - | - | - | - | - | - |

Table 2: Accuracy of attribute detection. Attribute detection using extracted witnesses versus using an existing layer in VGG-Face called *face descriptor* (i.e., the fc7 layer), whose neurons are considered representing abstract features of human faces [19].

| Dataset | VF [19] | | | | LFW [33] | | | |
|---|---|---|---|---|---|---|---|---|
| Attribute | Left Eye | Right Eye | Nose | Mouth | Left Eye | Right Eye | Nose | Mouth |
| Face Descriptor | 0.830 | 0.830 | 0.955 | 0.855 | 0.825 | 0.835 | 0.915 | 0.935 |
| Attribute Witness | 0.940 | 0.935 | 0.985 | 0.990 | 0.870 | 0.845 | 0.975 | 0.965 |

consistently achieve higher than 93% accuracy whereas face descriptor has lower than 86% for 3 attributes. Since LFW is a dataset with a different set of persons from those in the training set (from VF), we observe a decrease of accuracy. However, attribute witnesses still have higher accuracy than face descriptor. Hence, the results demonstrate that AmI extracts witnesses of high quality.

## 4.2   Detection of Adversarial Samples

We generate 100 adversarial samples for each attack discussed in §2. For patching attacks (Patch in Table 3), the code provided by Liu et al. [10] is utilized to generate adversarial samples. Since Sharif et al. [28] did not provide the attack code, we implemented their attack (Glasses) according to the paper. Three perturbation attacks (C&W$_0$, C&W$_2$ and C&W$_\infty$) are generated using the original implementation by Carlini and Wagner [8] . We use the CleverHans library [36] to generate untargeted attacks FGSM and BIM. For each targeted attack, we test on two settings similar to [18]: the target being the first output label (First) and the target being the next label of the correct one (Next). Note that successful attack samples include images that have overlapping attributes such as eyeglasses and beard. Although these overlapping attributes are not explicitly extracted, AmI can still successfully detect adversarial samples with their presence. From Table 3 it can be observed that AmI can effectively detect adversarial samples with more than 90% accuracy for most attacks. The same tests are conducted on feature squeezing (FS) [18]. The best result achieved by FS is 77%. FS especially does not preform well on FGSM and BIM, which is consistent with the observation of Xu et al. [18]. The results demonstrate that FS is not effective on FRSes in spite of its success on digit recognition (i.e., MNIST [37]) and object classification (i.e., CIFAR-10 [38]). We suspect that

Table 3: Detecting adversarial samples. We have two settings for targeted attacks. 'First' denotes that the first label is the target whereas 'Next' denotes the next label of the correct label is the target. FS stands feature squeezing; AS attribute substitution only; AP attribute preservation only; WKN non-witness weakening only; STN witness strengthening only; and AmI our final result. The bottom four rows denote detection accuracy of AmI using witnesses excluding some certain attribute (e.g., w/o Nose).

| Detector | FP | Targeted | | | | | | | | | | Untargeted | |
| | | Patch | | Glasses | | $C\&W_0$ | | $C\&W_2$ | | $C\&W_\infty$ | | FGSM | BIM |
| | | First | Next | First | Next | First | Next | First | Next | First | Next | | |
|---|---|---|---|---|---|---|---|---|---|---|---|---|---|
| FS [18] | 23.32% | 0.77 | 0.71 | 0.73 | 0.58 | 0.68 | 0.65 | 0.60 | 0.50 | 0.42 | 0.37 | 0.36 | 0.20 |
| AS | 20.41% | 0.96 | 0.98 | 0.97 | 0.97 | 0.93 | 0.99 | 0.99 | 1.00 | 0.96 | 1.00 | 0.85 | 0.76 |
| AP | 30.61% | 0.89 | 0.96 | 0.69 | 0.75 | 0.96 | 0.94 | 0.99 | 0.97 | 0.95 | 0.99 | 0.87 | 0.89 |
| WKN | 7.87% | 0.94 | 0.97 | 0.71 | 0.76 | 0.83 | 0.89 | 0.99 | 0.97 | 0.97 | 0.96 | 0.86 | 0.87 |
| STN | 2.33% | 0.08 | 0.19 | 0.16 | 0.19 | 0.90 | 0.94 | 0.97 | 1.00 | 0.76 | 0.87 | 0.46 | 0.41 |
| AmI | 9.91% | 0.97 | 0.98 | 0.85 | 0.85 | 0.91 | 0.95 | 0.99 | 0.99 | 0.97 | 1.00 | 0.91 | 0.90 |
| w/o Left Eye | 18.37% | 0.97 | 0.99 | 0.75 | 0.79 | 0.88 | 0.92 | 0.99 | 0.95 | 0.97 | 0.98 | 0.89 | 0.90 |
| w/o Right Eeye | 18.08% | 0.93 | 0.96 | 0.73 | 0.80 | 0.86 | 0.91 | 0.99 | 0.96 | 0.98 | 0.98 | 0.86 | 0.87 |
| w/o Nose | 27.41% | 0.97 | 0.99 | 0.78 | 0.84 | 0.91 | 0.94 | 0.98 | 0.97 | 0.99 | 0.98 | 0.94 | 0.90 |
| w/o Mouth | 20.99% | 0.91 | 0.97 | 0.74 | 0.79 | 0.86 | 0.95 | 1.00 | 0.95 | 0.99 | 0.98 | 0.86 | 0.87 |

feature squeezing only squeezes primitive features such as color depth so that it is less effective in complex tasks like face recognition that relies on more abstract features. However, it is unclear how to squeeze abstract features such as human eyes though.

We also evaluate the influence of the detection techniques on benign inputs. The false positive rate (the FP column in Table 3) is 9.91% for AmI and 23.32% for FS on the VF dataset. The same evaluation is also conducted on an external dataset, CelebA, which does not intersect with the original dataset. We randomly select 1000 images from CelebA and test on both FS and AmI. The false positive rate is 8.97% for AmI and 28.43% for FS. We suspect the higher false positive rate for FS is due to the same reason of squeezing primitive features that benign inputs also heavily rely on.

To support our design choice of using bi-directional reasoning in extracting attribute witnesses and using neuron strengthening and weakening, we conducted a few additional experiments: (1) we use only attribute substitution to extract witnesses and then build the attribute-steered model for detection with results shown in the AS row of Table 3; (2) we use only attribute preservation (AP); (3) we only weaken non-witnesses (WKN); and (4) we only strengthen witnesses (STN). Our results show that although AS or AP alone can detect adversarial samples with good accuracy, their false positive rates (see the FP column) are very high (i.e., 20.41% and 30.61%). The reason is that they extract too many neurons as the witnesses, which are strengthened in the new model, leading to wrong classification results for benign inputs. Moreover, using weakening has decent results but strengthening further improves them with a bit trade-off regarding false positives.

We further investigate the robustness of AmI employing different sets of witnesses by excluding those of some attribute during adversary detection. Specifically, we evaluate on four subsets of attribute witnesses extracted using bi-directional reasoning as shown in Table 3: (1) we exclude witnesses of left eye (w/o Left Eye); (2) exclude witnesses of right eye (w/o Right Eye); (3) exclude witnesses of nose (w/o Nose); and (4) exclude witnesses of mouth (w/o Mouth). We observe that the detection accuracy degrades a bit (less than 5% in most cases). It suggests that AmI is robust with respect to different attribute witnesses.

# 5   Conclusion

We propose an adversarial sample detection technique AmI in the context of face recognition, by leveraging interpretability of DNNs. Our technique features a number of critical design choices: novel bi-directional correspondence inference between face attributes and internal neurons; using attribute level mutation instead of pixel level mutation; and neuron strengthening and weakening. Results demonstrate that AmI is highly effective, superseding the state-of-the-art in the targeted context.

## 6  Acknowledgment

We thank the anonymous reviewers for their constructive comments. This research was supported, in part, by DARPA under contract FA8650-15-C-7562, NSF under awards 1748764 and 1409668, ONR under contracts N000141410468 and N000141712947, and Sandia National Lab under award 1701331. Any opinions, findings, and conclusions in this paper are those of the authors only and do not necessarily reflect the views of our sponsors.

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
