[Reviews · NeurIPS 2018]

Reviewer 1



The paper presented a novel way of detecting adversarial examples. Although the discussions were mainly centered around the specific domain of face recognition, I find the underlying ideas to be insightful and potentially more generally applicable to a wider range of tasks/DNN models as well. The efficacy of the identified witness neurons was also demonstrated through empirical studies and I think this in its own right makes it an interesting enough contribution. * That said, the biggest concern I have is on how much of this idea can be easily extended to other tasks because it seems the notion of this "attribute" is very much task-specific and, if not defined properly (e.g. being too broad or too specific), might either render the resulting attribute witness neurons non-existent or useless for interpretability purposes. Have the authors considered testing the robustness of this detection algorithm itself w.r.t. the set of "attributes" by conducting experiments with different sets of attributes (e.g. by adding/removing certain attribute from the set, comparing "attributes" across various granularities (e.g. from edgelets with different orientations to simply chunking the face images into upper/middle/lower regions, etc.) or types (e.g. geometry metrics of the face organs, etc.))? * How consistent were the identified witness neurons across the different inputs? And why just use 10 inputs (presumably the more the better)? * A slight notation issue: index i only appears on the l.h.s. of Eq.(7) and thus doesn't link well with anything on the r.h.s. * Please explain what you mean by "attribute conserving transformation" in more details.

Reviewer 2



In this paper the authors examine the intuition that interpretability to be the workhorse in detecting adversarial examples of different kinds. That is, if the humanly interpretable attributes are all the same for two images, then the prediction result should only be different if some non-interpretable neurons behave differently. Other than adversarial examples, this work is also highly related to interpretability and explainability questions for DNNs. The basis of their detection mechanism (AmI) lies in determining the sets of neurons (they call attribute witnesses) that are correspond (one-to-one) to a humanly interpretable attributes (like eyeglasses). That means, if the attribute does not change, the neuron should not give a different output, and the other way around if the feature changes, the neuron should change. Previously, given a neuron, people looked at a region with triggers the neuron the most (i.e. the latter). Authors argue that the former is also important and show that sets of neurons that can be identified uniquely (a subset of the set which satisfies one of the two) give a lower false positive rate. If only the latter criterion is satisfied, the neurons in question could in fact be spuriously correlated with the actual attribute (e.g. a stronger beard may change the color of that region and thus the neurons associated with the latter while it does not carry information about the characteristics of a beard). After detecting the attribute witnesses, the original trained prediction model is modified to form a new model, in which these attribute witness neurons are strenghtened, the rest is weakened. If a new input receives different predictions from original and new model, it is labeled as adversarial. The rationale behind this detection mechanism is that if the neurons for human interpretable attributes were activated in the original, then after strengthening these attributes and weakening the rest, the prediction result should not change and vice versa. In experiments on standard face datasets, they show that for binary attribute detection using activations of the attribute-witness neurons (for which an extra binary classifier is trained), their detection accuracy is much higher than just taking the last fully connected layer. This suggests these neurons do indeed carry information about the specific attribute. For adversarial sample detection, the true positive rate is much higher and the false discovery rate is much lower than "feature squeezing", a state-of-the-art technique. ----------------------------- Comments The paper is well-written. The aim, setup and contribution are clear. It is a unifying framework to detect adversarial attacks of various kinds. Some of the ideas such as checking consistency, receptive fields of a neuron had already been used by other works before which the authors cite, but the performance gain of identifying sets of "bi-directionally causal" sets of neurons with attributes has not been studied experimentally before. Their idea to use sounds intuitive and reasonable. One drawback/weakness of the empirical studies that I see is as follows: The attributes you labeled are just eyes, nose mouth which are locally separated. What would happen if e.g. eyeglasses and beard were to be included (just some more attributes from the CelebA dataset), which overlap? The rationale that adversarial attacks correspond to neurons that cannot be perceived by humans, the number of attributes should include locally-non separated ones. I can see that it would be hard to determine attribute witnesses for this (requiring substitutions and labeling by hand), which is what I would consider a major weakness of the method. Additionally of course the entire process of finding attribute witness requires very careful design as well as human labor (although it seems a rather small fraction of the dataset was needed for this stage, 10 if I understand correctly) and will highly depend on the domain if the technique were to be used in for other datasets. So another open question is: how do we determine the relevant (human interpretable) attributes for which we need to find the corresponding witnesses so that adversarial attacks can be detected? Furthermore, it would be nice if the authors could make the code publicly available. Minor comments: The minimal overlap between the attribute witnesses suggested by table 3 in the appendix is very interesting. Does that mean there really are neurons responding one attribute? Can you give some details about how the pairwise overlap is? It would also be nice to see attribute detection accuracy for the different AS, AP sets in table 2. I would avoid the word causality when talking about bi-directionality. I understand what you mean, but some people working in that field might not take it well. It is more of a unique mapping or correspondence between a set of neurons and an attribute. But there is no inherent causality between an input and a neuron in the statistical sense. ---------------------------- Details: non-relevant neurons are weakened to "suppress gut feeling" -why is this called gut feeling? The non-human interpretable feature probably will not trigger any "gut" either. For me "gut" is more related to intuition and interpretability than ... Might consider using a slightly different expression. Why does only strengthening witnesses not work at all for patch and glass attacks? I understand why AS and AP combined will give a lower false positive rate. But for strengthening vs. weakening shouldn't it somewhat "normalize" and only the ratio between witnesses and non-witnesses matter? I'm not quite sure where the asymmetry between weakening and strengthening effects come from.

Reviewer 3



1. Summary This paper proposes to identify adversarial face images for a NN by identifying which neurons change when facial features change and vice-versa. They make a new model by upweighting these neurons and downweighting the others. 2. High level paper I think the writing is a bit too informal. Despite this, the paper is mostly clear. I think the paper is a new take on how to deal with adversarial examples. There are sufficient experiments and I think this paper would be interesting to people, as long as it can be made scalable. As I am not an expert on adversarial methods I have lowered my confidence score. 3. High level technical One criticism I have is that the paper uses a lot of imprecise language. For this reason it's sometimes hard to figure out exactly what is going on, and it makes the paper less rigorous. Here are some specific examples: - Causality: Don't mention causality. Causality is a rigorous discipline with clearly defined assumptions and primitives. As soon as you mention causality the assumption should be that you are referring to the technical notion, which you aren't in this paper. I actually don't think you need to refer to causality, when you say you want 'forward causality' and 'backward causality', you really want some A to hold if and only if B holds, which is a logical biconditional, or if you want a mapping that maps exactly one element in A to one in B, this is a bijective mapping. - One-to-one: At Line 186, you mention a one-to-one mapping, but this is still a bit confusing because sometimes one-to-one functions describe injective functions (the forward half). So I would replace, wherever you mention causality with a logical biconditional if you're talking about two concepts A and B, and if you're talking about a mapping from things in A to things in B, I would use the word bijection. - Gut-feeling: I realize you are trying to make an analogy here but then either make it or don't make it. Specifically, if you are trying to make a claim that this is similar to 'gut-feeling' in humans then you should make this more precise. My preference would be to just use a different phrase such as 'human-uninterpretable'. By the same token, you could replace 'reasoning aspect' (line 67) with 'human-interpretable portion'. 4. Low level technical - Line 43: what does this mean: 'That is, the DNN is guessing for adversarial samples'? - Line 162: 'This is analogous to the retrospecting...' there should definitely be a citation for this - Line 191: 'equivalent' -> 'logically equivalent' 5. 1/2 sentence summary Overall, despite some loose wording I think this paper makes a nice contribution. Post-Rebuttal ----------------------- As my main issue with the paper is its loose wording and the authors agree to make the changes I suggest, I am happy to keep my score the same.